# The Creation of True Two-Dimensional Silicon Carbide

**DOI:** 10.3390/nano11071799

**Published:** 2021-07-10

**Authors:** Sakineh Chabi, Zeynel Guler, Adrian J. Brearley, Angelica D. Benavidez, Ting Shan Luk

**Affiliations:** 1Department of Mechanical Engineering, University of New Mexico, Albuquerque, NM 87131, USA; zeynelguler@unm.edu; 2Department of Earth and Planetary Sciences, University of New Mexico, Albuquerque, NM 87131, USA; brearley@unm.edu; 3Center for Microengineered Materials, Department of Chemical and Biological Engineering, University of New Mexico, Albuquerque, NM 87131, USA; asanch18@unm.edu; 4Center for Integrated Nanotechnologies, Sandia National Laboratories, Albuquerque, NM 87123, USA; tsluk@sandia.gov

**Keywords:** two-dimensional materials, silicon carbide, siligraphene, semiconductors, graphene

## Abstract

This paper reports the successful synthesis of true two-dimensional silicon carbide using a top-down synthesis approach. Theoretical studies have predicted that 2D SiC has a stable planar structure and is a direct band gap semiconducting material. Experimentally, however, the growth of 2D SiC has challenged scientists for decades because bulk silicon carbide is not a van der Waals layered material. Adjacent atoms of SiC bond together via covalent sp^3^ hybridization, which is much stronger than van der Waals bonding in layered materials. Additionally, bulk SiC exists in more than 250 polytypes, further complicating the synthesis process, and making the selection of the SiC precursor polytype extremely important. This work demonstrates, for the first time, the successful isolation of 2D SiC from hexagonal SiC via a wet exfoliation method. Unlike many other 2D materials such as silicene that suffer from environmental instability, the created 2D SiC nanosheets are environmentally stable, and show no sign of degradation. 2D SiC also shows interesting Raman behavior, different from that of the bulk SiC. Our results suggest a strong correlation between the thickness of the nanosheets and the intensity of the longitudinal optical (LO) Raman mode. Furthermore, the created 2D SiC shows visible-light emission, indicating its potential applications for light-emitting devices and integrated microelectronics circuits. We anticipate that this work will cause disruptive impact across various technological fields, ranging from optoelectronics and spintronics to electronics and energy applications.

## 1. Introduction

Two-dimensional silicon carbide has received significant attention recently, and the structure and fundamental properties of 2D SiC and related materials have been investigated by various theoretical studies. This rapidly increasing interest comes from the immense potential and promise that such material holds for the future. As a wide-bandgap semiconducting material with high thermal capability, SiC is a key material for various technological fields ranging from high-power electronics and photonic devices to high-temperature devices and quantum information processing. The 2D form of SiC will naturally benefit from these overall SiC properties. Furthermore, as a result of reduced dimensionality and quantum confinement, 2D SiC is predicted to exhibit exotic optical and electronic properties, that are very useful for various applications [1,2,3,4,5]. Structurally, 2D SiC is predicted to have a graphene-like honeycomb structure consisting of alternating Si and C atoms. In the monolayer SiC, the C and Si atoms bond through sp^2^ hybridization to form the SiC sheet, Figure 1a.

One of the most fascinating features about monolayer SiC is that it has a direct wide-bandgap, Figure 1b. This feature will address countless drawbacks associated with the gapless nature of graphene and the indirect band gap of bulk SiC, Figure 1c. The band gap opening in 2D SiC is related to the electronegativity differences between silicon and carbon atoms, which induces electron transfer from valence electrons of Si to the nearest C, causing a band gap to emerge [6,7,8]. Based on density functional theory, monolayer SiC has a direct band gap of about 2.58 eV [9,10,11,12,13]. The indirect–direct band gap transition characteristic in 2D SiC is similar to the previously reported feature in other 2D materials, such as 2D transition metal dichalcogenides (TMDs).

Theoretical studies have also found that 2D SiC has very rich optical properties, such as strong photoluminescence, non-linear optical properties, and excitonic effects as a result of quantum confinement effects [4,14,15,16]. In terms of mechanical properties, 2D SiC is one of the toughest and stiffest 2D materials. Only graphene and h-BN are stiffer than monolayer SiC [10,17].

The thermodynamic and kinetic stability of bilayer and multilayer SiC has also been investigated. Various theoretical studies have found that only monolayer SiC has a stable graphene-like structure. Multilayer SiC and even bilayer SiC are unable to form planar graphene- like structures. Multilayer SiC resembles the bulk SiC in their structure and properties. Like bulk SiC, both multilayer and bilayer SiC have indirect band gap [18,19,20]. A detailed discussion about the structure, properties, and applications of 2D SiC has been provided in our very recent review paper [5].

Although the concept of sp^2^ bonding in 2D SiC might look challenging, as opposed to the naturally occurring sp^3^ bonding in bulk SiC, all theoretical studies have confirmed that 2D SiC is 100% planar [14,18,19,21,22,23,24]. Using first-principle calculations (DFT), Freeman et al. [19] predicted that as wurtzite structures, such as ZnO and 6H-SiC, become only a few atoms thick, they adopt a graphitic structure, as it is the most stable structure for these ultrathin materials. In fact, 2D SiC is not the first structure to contain Si=C bonding. A variety of Si=C containing compounds, known as “silenes”, have been reported in the past [25,26]. Thus, monolayer SiC can stabilize itself by adopting a planar, sp^2^ structure. However, given that bulk SiC has a tetrahedral sp^3^ structure, the phase transformation from sp^3^ to sp^2^, Figure 1d–e, must take place if a top-down approach is going to be used for the isolation of 2D SiC. The Si-C bond length in 2D SiC is predicted to be 1.79 A^°^, which is shorter than 1.89 A^°^ found in bulk SiC. The predicted lattice constant, a, for monolayer SiC is 3.1 A^°^, which is larger than that in bulk SiC, as shown in Figure 1f [18,19,27,28,29,30]. Table 1 lists structural characteristics of 2D SiC and related materials.

In addition to 2D SiC, other compositions of silicon carbide, i.e., 2D SixCy, are also energetically favorable. Thermodynamic and kinetic stability of 2D SixCy compositions have also been investigated [3,23,32,33,34,35,36,37]. All theoretical studies confirm that 1:1 stoichiometry, i.e., SiC, is the most stable composition.

Unlike other 2D materials, such as silicene, the 2D analog of silicon, or few-layer black phosphorus (BP) that stabilize themselves through buckling, zero buckling is predicted for 2D SiC [22,40,41]. Thus, indeed like graphene and h-BN, monolayer SiC has a stable planar structure. Practically, however, there are only a few published experimental works on multilayer SiC nanosheets [42,43,44]. These previous works used chemical vapor deposition, chemical exfoliation, and hydrothermal methods to prepares SiC nanosheets. However, a graphene-like SiC structure has not been reported yet. Here we report the first successful synthesis of graphene-like 2D SiC via a wet exfoliation process.

## 2. Methods and Materials

To form 2D SiC, hexagonal bulk SiC (BeanTown Chemical, Hudson, NH, USA) was exfoliated in isopropyl alcohol or N-methyl-2- pyrrolidone (Sigma-Aldrich, St. Louis, MO, USA), using a bath sonication for 24 h (Branson Ultrasonic Corporation, Denbury, CT, USA). The average concentration of the SiC dispersion is 1 mg/mL. The dispersion was then centrifuged at an average rate of 1000 rpm for about 5 min (Eppendorf AG, Hamburg, Germany). TEM images were recorded using STEM and HRTEM mode (JEOL 2010F operating at 200 kV) with the samples on top of a holey carbon grid. The Raman spectra were collected with an excitation laser beam of 532 nm (WITec Alpha300R). 

For most of the characterization tests, e.g., Raman, XRD, SEM, AFM, and PL, drops of a SiC dispersion were placed on different substrates such as a holey carbon grid or silicon substrate, and then dried at ambient conditions prior to the characterization. SEM images were collected using a FEI Quanta 3D FEGSEM instrument. XRD tests were performed using a Rigaku SmartLab instrument (Rigaku, Japan) equipped with Cu KÁ radiation. The AFM measurements were conducted in the tapping mode using a Dimension 3100 instrument (Veeco) and a silicon tip (NSC15/AL BS, Micromasch). PL measurements were performed at room temperature, using a UV micro-photoluminescence system and a sapphire substrate. PI samples were excited by a 269 nm excitation source. Optical microscopy images were recorded using a BX53MRF-S Olympus microscope, (Olympus, Japan). 

Absorption spectra were obtained using a UV-2600i UV-Vis Shimadzu spectrophotometer (Shimadzu Corp, Japan). To obtain the absorption spectra, the samples from the supernatant were placed in quartz cuvettes and measured against a reference sample with only the respective solvent in the cuvette.

## 3. Results and Discussion

Silicon carbide nanosheets were produced via wet exfoliation of bulk hexagonal SiC in isopropyl alcohol (IPA) or N-methyl-2-pyrrolidone (NMP) solvents, Figure 2a. The exfoliation process consists of two steps: (i) wet exfoliation of SiC precursor via bath sonication and (ii) centrifugation. Throughout the sonication process, the bath temperature was kept below 50 °C, Appendix A. After the completion of the centrifuge step, the samples were transferred onto SiO2/Si or other substrates for various characterization. The thickness of the SiO2 layer is 300 nm. Figure 2 shows the results from optical microscopy, atomic force microscopy, AFM, scanning electron microscopy, SEM, and measurements. The optical image, Figure 2b, shows nanosheets with different sizes in the range of 200 nm–2 μm. Appendix A presents more optical microscopy images.

Figure 2c shows AFM images of two SiC nanosheets. The height profiles of these nanosheets are presented in Figure 2d. As shown, the average height of the red and blue flakes is 0.28 and 0.25 nm, respectively. Thus, we can conclude that the shown nanosheets in Figure 2c are monolayers, as the interlayer spacing in bulk SiC is about 0.25 nm (the dominant plane). Appendix A presents the related histograms. Figure 2e–h shows an SEM image of another SiC nanosheet, and the associated elemental analysis. The chemical composition and purity of the liquid exfoliated nanosheets is confirmed by X-ray elemental mapping by SEM. As shown in Figure 2g,h, the produced nanosheets are 100% silicon carbide. Only silicon and carbon were observed in the elemental maps.

Figure 3 presents the results from transmission electron microscopy, TEM, imaging. Figure 3a, b shows the overall morphology of the SiC nanosheets. The contrast variation within the nanosheets could be related to the thickness change, or folded edges. The composition of the SiC flakes was initially confirmed by EDX measurements, as shown in Figure 3c. As can be seen from the EDX spectrum, the sample in Figure 3b only contains Si and C, with an atomic ratio of ~1, thus the produced nanosheets are pure silicon carbide. Figure 3d is a high-resolution TEM image of the created 2D SiC. This HRTEM image clearly indicates the highly ordered crystalline structure of the exfoliated SiC nanosheets. All HRTEM images, including those from the edges of the nanosheets, revealed that the created nanosheets have a highly crystalline structure. Figure 3e is a magnified image of the circled region shown in Figure 3d. The graphene-like hexagonal structures shown here in Figure 3e confirm the creation of 2D SiC. Figure 3f is a magnified image of the hexagonal structure shown in Figure 3e. This image further confirms the creation of stable graphene-like 2D SiC in this study, as the honeycomb structure is easily observed in these HRTEM images.

We then calculated the lattice constant of the created 2D SiC using the ImageJ software [45] (ImageJ, US National Institutes of Health). Figure 3g shows the relative profile intensity for 2D SiC shown in Figure 3e (along the green line). We obtained an average lattice constant of 3.1 ± 0.01 Å. This value agrees well with theoretical prediction and further confirms that the SiC nature of these materials. The height similarity of the peaks in Figure 3g indicates similar electron densities in the hexagonal rings, which further confirms the monolayer nature of the shown structure.

Figure 3h is a high-resolution TEM image of another 2D SiC sample, and Figure 3i is a magnified picture of the circled region in Figure 3h. These images are very interesting, as they show an extended graphene-like SiC lattice. Although some portions of the hexagonal lattice in this image look distorted because of the light contrast, there is still one lattice structure. These results confirm that these images indeed belong to monolayer SiC. Appendix A shows an intensity profile for the sample shown in Figure 3i. The intensity variation of the line profile further reveals that the nanosheet shown in Figure 3i belongs to monolayer SiC. Only monolayer SiC sheets show such a significant variation in intensity between neighboring atoms, without any contribution from the background. Appendix A shows more TEM images from the exfoliated SiC nanosheets.

It is worth emphasizing that since bulk SiC, including the 6H SiC precursor, has well known polytype crystal structures, HRTEM can be used efficiently to differentiate between single layer SiC and multilayer or bulk SiC. For example, 6H SiC has ABCACB stacking sequences, 2H SiC has AB order, or 3C-SiC has ABC stacking structure. In all these types, the second layer will be arranged differently than the top layer, and there will be contrast differences in their TEM images. For example, in the AB stacking sequence, the second layer (along the c-axis) is shifted parallel with respect to the first one. Thus, the absence of these stacking sequences or layer contrast in our TEM results further confirms the successful isolation of 2D SiC.

Figure 4a,b show and compares the Raman results of the SiC precursor, powder, and exfoliated SiC nanosheets. In the case of the SiC precursor, the Raman modes observed at 780, 790, and 970 cm^−1^ are consistent with the Raman modes of 6H-SiC. The peak around 790 cm^−1^ is the characteristic peak of silicon carbide and corresponds to the transverse optical (TO) phonon vibrational mode of the Si–C bond. The other peak around 780 cm^−1^ belongs to the TO mode of the SiC bond. The peak around 970 cm^−1^ can be indexed to the longitudinal optical (LO) of A1 phonon in silicon carbide. Figure 4b shows the Raman spectrum of the 2D SiC.

Appendix A shows Raman spectrum of 2D SiC in a wider range (600–3000 cm^−1^). This spectrum further confirms the SiC nature of the created nanosheets, as both peaks belong to silicon carbide.

Similar to the precursor, 2D SiC shows an intense TO peak around 790 cm^−1^, which is characteristic of all SiC materials, and also one LO peak. Despite these similarities, there are significant differences between the Raman spectra of SiC precursor and 2D SiC. As can be seen from Figure 4b, 2D SiC has a much weaker and broader LO peak, around 930 cm^−1^, than the SiC precursor. This asymmetric broadening and down shifting of the LO peak in 2D SiC is a direct result of quantum confinement effects. Furthermore, the disappearance of the TO peak in the low frequency regime, 780 cm^−1^, the shoulder, reveals some structural changes during the exfoliation process of 6H-SiC. These changes are a direct result of the evolution/transition of electronic bands and phonons with the number of layers. The weakening of the LO mode in 2D SiC could be related to the surface conversion from sp [3] to sp [2] hybridization. However, it is important to emphasize that Raman measurements were performed on a drop-cased dispersion of SiC nanosheets, and not on pure monolayer SiC. As such, a full understanding of these transitions requires comprehensive characterization tests of both monolayer and few-layers SiC. This will be investigated in our next paper.

We also used X-ray diffraction (XRD), to further characterize the exfoliated nanosheets, as shown in Appendix A. When only 10 or even 20 drops of the dispersion were placed on the substrate, no peak was observed in the XRD scans. The XRD scan shown in Appendix A was collected using parallel beam optics at a grazing angle of 1 degree. In this case, more than 50 drops were deposited on a silicon substrate. Thus, the sample is not pure 2D SiC anymore. Only one very weak and broad peak, which belongs to SiC, was observed around 35.8 degrees. This peak reveals a *d*-spacing of about 0.25 nm which agrees well with the interlayer spacing in silicon carbide crystallographic planes. The absence of peaks in the XRD scans is related to the atomic thickness of the produced nanosheets. Only stacked thick films can show intense, sharp peaks in XRD.

Thus, as our TEM, and Raman results showed, 2D SiC is attainable. It can be realized via a wet exfoliation process of 6H-SiC. The successful exfoliation of SiC nanosheets from bulk SiC could be related to the solvents used in this study and their efficient interaction (both physically and chemically) with the SiC precursor. In fact, parameters such as bulkiness of the NMP solvent and i its high surface tension, or hydrogen bonding in IPA and its ability to modify the charge carrier density in materials (via acting as an electron acceptor), could be one of the main driving forces behind the successful isolation of 2D SiC [46,47,48]. Earlier studies on silicon/carbon double bonds showed that the sp^3^ to sp^2^ transitions in Si=C containing materials could be stabilized via mechanisms such as surface depolarization, electronic effects, and steric protection by large substitutes [25,26,49,50,51]. Thus, both NMP and IPA could contribute positively, and even initiate such mechanisms. However, a comprehensive understanding of the isolation mechanism demands more studies.

Figure 4c presents the absorbance spectrum from solution dispersion of 2D SiC. As shown in Figure 4c, the created SiC nanosheets absorb photons at visible light range. The absorption spectrum has two peaks around 2.14 and 2.58 eV, and one small peak around 2.3 eV, corresponding to π→π^∗^ transitions. Our absorption data from various solutions of 2D SiC dispersions (in the visible region), reveal that all tested samples have discrete peaks in the range of 2.13–2.7 eV, with the 2.13 eV peak being the most pronounced peak in the visible region. The formation of these discrete peaks could be attributed to the strongly-bound exciton in 2D SiC, quantum confinement effects, and even surface defects. Since we tested solution dispersions of SiC nanosheets, as opposed to pure monolayer SiC, more work needs to be conducted before identifying one specific electronic structure/transition or process to each peak.

Figure 4d shows the photoluminescence, PL, result from drop casted SiC nanosheets on a sapphire substrate. The excitation wavelength was 266 nm. Our experimental results, from different 2D SiC samples, showed that, unlike bulk SiC, 2D SiC has strong PL properties. Bulk SiC is known to have poor PL properties. As shown in Figure 4d, 2D SiC has a visible emission band at about 2.58 eV (or 480 nm). The good match between the position of the PL peak and one of the absorption peaks (at 2.58 eV) is very interesting and it is a strong indication of a direct band gap transition in the created 2D SiC nanosheets. These results indicate that 2D SiC may be used for blue-green luminescent devices, e.g., light emitting diodes, as well as integrated micro/nano electronic circuits, such as LED integrated computer chips and biolabeling and biosensing. Our absorption and PL data from other samples suggest that the position and intensity of the PL peaks is also affected by the substrate, excitation wavelength, solvent types, and synthesis parameters. Ideally, PL tests should be performed on suspended pure monolayer SiC samples. A detailed and comprehensive analysis of PL performance of 2D SiC is beyond the scope of this paper.

## 4. Conclusions

In conclusion, we have performed the first successful exfoliation of true 2D SiC from bulk SiC. Our TEM and Raman results showed that monolayer SiC has a stable graphene-like structure, and that the produced nanosheets are of high purity and crystallinity. We have also analyzed the optical properties of SiC nanosheets. The results showed that the nanosheets have strong emission in the visible range. With a direct band gap at the monolayer limit, 2D SiC represents an interesting platform for the next generation of electronics and optoelectronics technologies.

Unlike bulk SiC, which exists in more than 250 polytypes, monolayer SiC does not have any polytype. Thus, the application of 2D SiC would be less complicated than existing SiC materials. We envision that successive breakthrough in 2D SiC and related SixCy materials could usher in a new era of semiconducting materials with exciting applications in optoelectronics and electronics, bioimaging and sensing, and computing.

## Figures and Tables

**Figure 1 nanomaterials-11-01799-f001:**
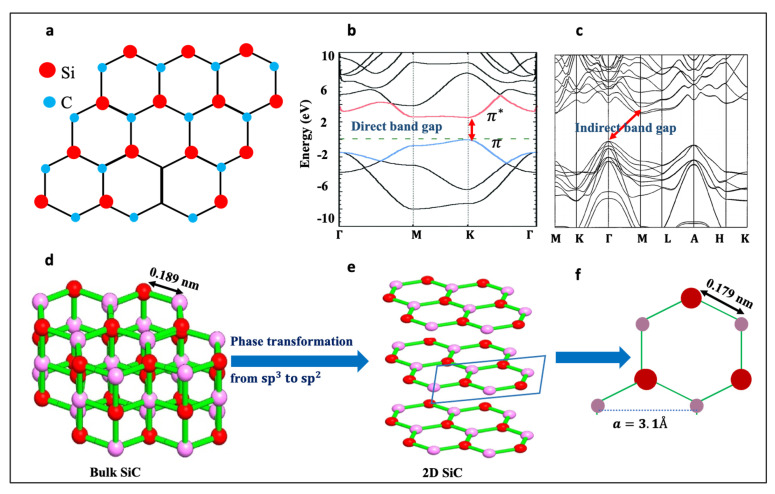
(**a**) The chemical structure of monolayer SiC, (**b**) Electronic band structures of 2D SiC, and bulk 6H-SiC, respectively. Republished from [22] with the permission of RSC. Figure 1c. (**c**) Electronic band structures of 6H-SiC. Reproduced from [31] with the permission of AIP publishing. (**d**,**e**) a schematic of phase transformation in 2D SiC. (**f**) a schematic of the unit cell of 2D SiC, showing the in-plane lattice constant, a, and atomic bonding of 2D SiC.

**Figure 2 nanomaterials-11-01799-f002:**
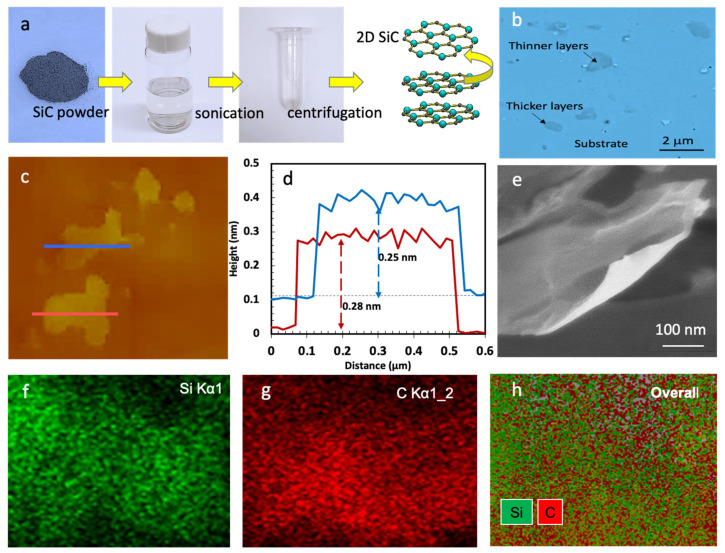
(**a**) Photographs of the exfoliation process and the dispersed SiC, (**b**) optical microscopy image, (**c**) AFM images, (**d**) height profiles of the SiC nanosheets marked in c, (**e**) SEM image of SiC nanosheets and (**f**–**h**) elemental X-ray maps for Si, C, and an RG (Si, C) composite image of the shown SiC nanosheet in Figure 2e.

**Figure 3 nanomaterials-11-01799-f003:**
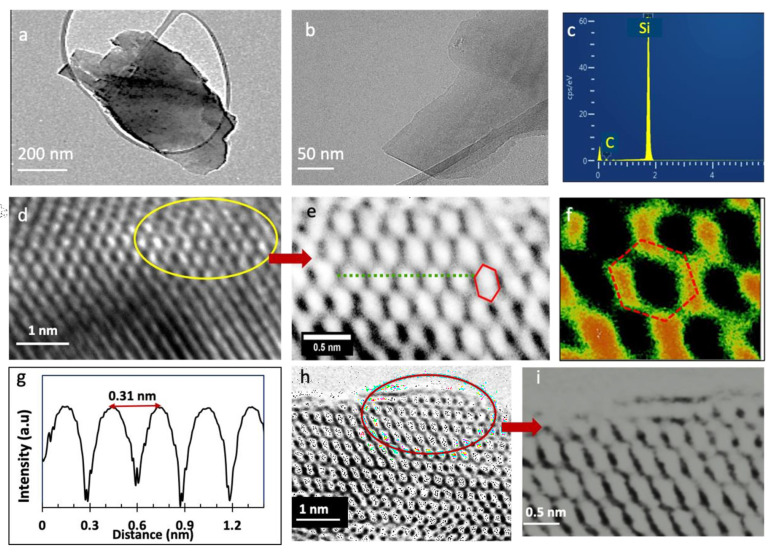
TEM characterization of exfoliated SiC nanosheets. (**a,b**) Low magnification bright-field TEM images of SiC nanosheets. (**c**) EDX analysis (by TEM) of the nanosheet shown in (**b**). (**d**–**f,h,i**) high- resolution TEM images of the ordered structure of the SiC nanosheets. (**g**): Intensity distributions along dotted lines shown in Figure 3h.

**Figure 4 nanomaterials-11-01799-f004:**
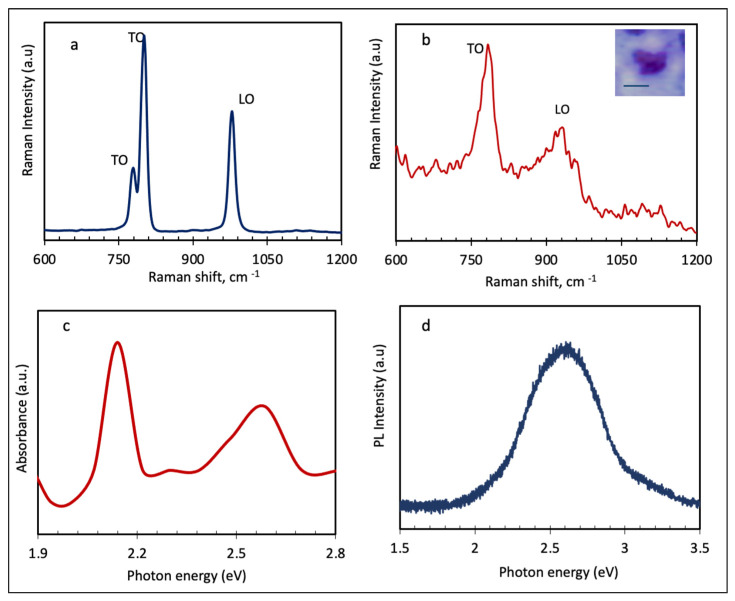
(**a**,**b**) Raman spectra of SiC precursor and exfoliated SiC nanosheets, respectively. The inset of image b shows an optical image of the tested SiC nanosheets prior to the measurement. The scale bar in the inset of 4b is 500 nm. (**c**) optical absorption spectrum of 2D SiC dispersion. (**d**) Photoluminescence spectrum of drop-casted SiC nanosheets on a sapphire substrate.

**Table 1 nanomaterials-11-01799-t001:** Structural characteristics of 2D SiC and related materials.

Material	Bond Length (Å)	Lattice Constant (Å)	Bonding	Band Gap (eV)	References
2D SiC	1.77–1.79	3.1	Planar- sp^2^	Direct: 2.58	[8,18]
6H-SiC	1.89	3.08	Hexagonal-sp^3^	Indirect: 3.2	[18,30]
3C-SiC	1.89	4.3	Cubic- sp^3^	Indirect: 2.2	[38]
Graphene	1.42	2.46	Planar-sp^2^	Zero	[8,18]
Silicene	2.27	3.8	Buckled	Zero	[3,10,39]

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
