# Peer review of "The Creation of True Two-Dimensional Silicon Carbide"

_nanomaterials, 2021, doi:10.3390/nano11071799_

Round 1

Reviewer 1 Report

In the revised paper, the authors have responded to some of my comments from the initial referee report. As far as I see, Raman and TEM seem to provide the strongest arguments for claiming that the layers are really single layers. All other results are could also be explained by the formation multilayers. However, a number of points still have to be taken care of until the paper might be ready for publication. 

  1. Figure 1: Figure 1b seems to be bad, the x-axis of Figure 1c is partly obstructed by Figure 1f.
  2. page 7, line 1: what is the thickness of the SiO2 layer? What optical contrast is expected for SiC single layer (see, e.g., Blake, P.; Hill, E.; Castro Neto, A.; Novoselov, K.; Jiang, D.; Yang, R.; Booth, T. & Geim, A., Making graphene visible, Appl. Phys. Lett., 2007, 91, 063124)
  3. page 9: the authors should add a reference for ImageJ such as Schneider, C. A.; Rasband, W. S. & Eliceiri, K. W. NIH Image to ImageJ: 25 years of image analysis, Nat. Methods, 2012, 9, 671-675.
  4. page 9, line 3: it is rather (3.1 +/- 0.01) Angstrom instead of 3.1 Angstrom +/- 0.01
  5. The authors should be more careful providing correct references. For example, refs. 4, 11, and 13 are lacking either volume or page numbers. Refs. 25 and 26 are lacking the journal names. For ref 26 part of the paper text is given. In ref. 33 the title has a typo ("SiC_2", not "SiC 2"). Similar for ref 37. All references have to be rechecked!
  6. Figure S4b: why is the unit for the x-axis "a.u." and not "nm"?

Author Response

Referee: 1

Comments and Suggestions for Authors

In the revised paper, the authors have responded to some of my comments from the initial referee report. As far as I see, Raman and TEM seem to provide the strongest arguments for claiming that the layers are really single layers. All other results are could also be explained by the formation multilayers. However, a number of points still have to be taken care of until the paper might be ready for publication. 

  1. Figure 1: Figure 1b seems to be bad, the x-axis of Figure 1c is partly obstructed by Figure 1f.

RE: Thanks for the comment. We fixed both graphs in the revised manuscript. New changes are highlighted in Figure 1.

2. page 7, line 1: what is the thickness of the SiO2 layer? What optical contrast is expected for SiC single layer (see, e.g., Blake, P.; Hill, E.; Castro Neto, A.; Novoselov, K.; Jiang, D.; Yang, R.; Booth, T. & Geim, A., Making graphene visible, Appl. Phys. Lett., 2007, 91, 063124)

RE: It is 300 nm. We added this info in page 7. The new changes are highlighted. Regarding the optical contrast, we did not investigate it. We will look into it in our next paper.

3. page 9: the authors should add a reference for ImageJ such as Schneider, C. A.; Rasband, W. S. & Eliceiri, K. W. NIH Image to ImageJ: 25 years of image analysis, Nat. Methods, 2012, 9, 671-675.

RE: Done

4. page 9, line 3: it is rather (3.1 +/- 0.01) Angstrom instead of 3.1 Angstrom +/- 0.01

RE: Done

5. The authors should be more careful providing correct references. For example, refs. 4, 11, and 13 are lacking either volume or page numbers. Refs. 25 and 26 are lacking the journal names. For ref 26 part of the paper text is given. In ref. 33 the title has a typo ("SiC_2", not "SiC 2"). Similar for ref 37. All references have to be rechecked!

RE: Thanks for the comment. We fixed them in the revised manuscript.

6.Figure S4b: why is the unit for the x-axis "a.u." and not "nm"?

RE: It is “pixels”.

Reviewer 2 Report

I am satisfied by the answers on the raised issues. In the present version the MS diserverse acceptance as is. 

Author Response

Thank you very much !